# A deep learning framework for predicting endometrial cancer from cytopathologic images with different staining styles

**Ruijie Wang**[1], **Qing Li**[2], **Guizhi Shi**[3], **Qiling Li**[2], **Dexing Zhong**[1,4,5]*

**1** School of Automation Science and Engineering, Xi'an Jiaotong University, Xi'an, Shaanxi, P.R. China, **2** Department of Obstetrics and Gynecology, The First Affiliated Hospital of Xi'an Jiaotong University, Xi'an, Shaanxi, P.R. China, **3** Laboratory Animal Center, Institute of Biophysics, Chinese Academy of Sciences, and the University of Chinese Academy of Sciences, Beijing, China, **4** Pazhou Laboratory, Guangzhou, P.R. China, **5** Research Institute of Xi'an Jiaotong University, Zhejiang, Hangzhou, P.R. China

* bell@xjtu.edu.cn

**Data Availability Statement:** Data cannot be shared publicly because of privacy protection. The owner of the data is the Ethics Committee of the First Affiliated Hospital of Xi'an Jiaotong University,

## Abstract

Endometrial cancer screening is crucial for clinical treatment. Currently, cytopathologists analyze cytopathology images is considered a popular screening method, but manual diagnosis is time-consuming and laborious. Deep learning can provide objective guidance efficiency. But endometrial cytopathology images often come from different medical centers with different staining styles. It decreases the generalization ability of deep learning models in cytopathology images analysis, leading to poor performance. This study presents a robust automated screening framework for endometrial cancer that can be applied to cytopathology images with different staining styles, and provide an objective diagnostic reference for cytopathologists, thus contributing to clinical treatment. We collected and built the XJTU-EC dataset, the first cytopathology dataset that includes segmentation and classification labels. And we propose an efficient two-stage framework for adapting different staining style images, and screening endometrial cancer at the cellular level. Specifically, in the first stage, a novel CM-UNet is utilized to segment cell clumps, with a channel attention (CA) module and a multi-level semantic supervision (MSS) module. It can ignore staining variance and focus on extracting semantic information for segmentation. In the second stage, we propose a robust and effective classification algorithm based on contrastive learning, ECRNet. By momentum-based updating and adding labeled memory banks, it can reduce most of the false negative results. On the XJTU-EC dataset, CM-UNet achieves an excellent segmentation performance, and ECRNet obtains an accuracy of 98.50%, a precision of 99.32% and a sensitivity of 97.67% on the test set, which outperforms other competitive classical models. Our method robustly predicts endometrial cancer on cytopathologic images with different staining styles, which will further advance research in endometrial cancer screening and provide early diagnosis for patients. The code will be available on GitHub.

and therefore it is not freely available. The Ethics Committee of the First Affiliated Hospital of Xi'an Jiaotong University imposed these restrictions. Data are available from the Ethics Committee (xjyfyllh@163.com) for researchers who meet the criteria for access to confidential data. The externally validated data were obtained from the public data platform, AIstudio, accessible via the Internet (https://aistudio.baidu.com/datasetdetail/273988).Anyone can access this data after registering an account on this platform.

**Funding:** This work is supported by National Natural Science Foundation of China (No. 62376211; 62206218), Natural Science Foundation of Zhejiang Province (No. LTGG23F030006), Special Project for Technological Innovation Guidance of Shaanxi Province (No. 2024ZC-YYDP-24).

**Competing interests:** The authors have declared that no competing interests exist.

## Introduction

Endometrial cancer is one of the most common tumors in the female reproductive system and usually occurs in postmenopausal women [1, 2]. It is the leading cause of cancer-related deaths in women worldwide [3], with approximately 76,000 deaths each year [1]. And the incidence and mortality of endometrial cancer is expected to continue to rise in the coming decades [4, 5]. Studies have shown that endometrial screening can help to detect cellular lesions, and improve long-term patient outcomes [6, 7]. It would significantly improve survival rates [8]. So, endometrial cancer screening is crucial.

However, there are few tools available for the endometrial cancer screening. A minimally invasive method based on cytopathology to address endometrial cancer screening is a hot topic of current research and future development [9]. And it has been widely used in countries such as Japan [10, 11]. Moreover, it is considered to be cost-effective and more useful for early screening than invasive endometrial biopsy and hysteroscopy [12–15]. Nevertheless, there are still many difficulties in advancing cytopathological screening.

Firstly, there are no endometrial cytopathology datasets that contain segmentation and classification labels, due to the difficulty of data acquisition and the high cost of high-quality annotation. To combat the challenge, our team collecting 139 cytopathology whole slide images (WSI) with our own designed endometrial sampling device Li Brush (20152660054, Xi'an Meijiajia Medical Technology Co., Ltd., China). Among them, 39 WSIs are papanicolaou stained, and 100 WSIs are hematoxylin and eosin (H&E) stained. These WSIs are annotated by two cytopathologists, thus building a dataset for cytological screening of endometrial cancer. To the best of our knowledge, this is the first cytopathology dataset that includes segmentation and classification labels.

Secondly, diagnosing cytopathological slides is a time-consuming and complex task [16]. Subjective discrepancies and heavy workloads affect the productivity of cytopathologists [17]. As a powerful tool, deep learning can provide objective references for doctors and further improve their work efficiency [18, 19]. Therefore, it is widely used in thyroid cancer [20], cervical cancer [21], liver cancer [22, 23], and other diseases to improve the diagnostic efficiency [24]. And in endometrial diagnosis, deep learning is usually used for segmentation and classification tasks. In the field of segmentation, Erlend Hodneland et al. used a UNet-based 3D convolutional neural network (CNN) to segment endometrial tumors on radiology images, which aimed at generating tumor models for better individualized treatment strategies [25]. Zhiyong Xia et al. designed a dense pyramidal attention U-Net for hysteroscopic images and ultrasound images, which can help doctors to accurately localize the lesion site [26]. In addition, segmentation algorithms are often used to assist in confirming the depth of myometrial infiltration in endometrial cancer [27–30].In the field of classification, Christina Fell et al. used CNNs to classify endometrial histopathology images at the WSI level, which are categorized as "malignant", "other or benign" and "insufficient" [31]. Sarah Fremond et al. proposed an interpretable endometrial cancer classification system, which can further predict four molecular subtypes of endometrial cancer through self-supervised learning [32]. Min Feng et al. develop a deep learning model for predicting lymph node metastasis from histopathologic images of endometrial cancer, which is believed to predict metastatic status and improve accuracy [33]. In summary, In summary, we note that deep learning models are commonly used for radiology images [34, 35] and histopathology images [36–38] in endometrial studies. Therefore, there is still a lack of endometrial cancer screening algorithms based on cytopathologic images. The aim of our study is to develop a new algorithm that learns cytopathological features and provides an assisted diagnostic strategy.

Here we propose an innovative two-stage framework for endometrial cancer screening. In this study, we found that the staining styles of slides was performed differently in different medical centers [39]. Some endometrial samples were stained with H&E, while others were stained with papanicolaou. In addition, the stained slides can also be highly variable due to the preservation environment, changes in the scanner, etc [39]. This can affect the final diagnosis results [40, 41]. Therefore, we have improved the automated screening framework to increase its robustness and accuracy.

In clinical diagnosis, cell clumps are regions of interest (ROIs) for cytopathologists, while the background contains unnecessary noises [42]. So, in the first stage, we propose an improved segmentation network CM-UNet, which extracts ROIs from cytopathology images. We introduce a channel attention (CA) module and a multilevel semantic supervision (MSS) module to obtain more local and global contextual representations. In addition, we added novel skip connections to efficiently extract multi-scale features.

In the second stage, we need to classify ROIs to screen positive cell clumps. Since the obtained ROIs vary in shape and size, the different representations among these ROIs may affect the performance of the classification model. We propose the contrastive learning based algorithm ECRNet to classify ROIs. In contrast to current contrastive learning methods that treat different augmentations of the same image as positive pairs, we introduce the label memory bank to preserve the representation information of the image and the corresponding labels. ECRNet treats two instances with the same label as a positive sample pair, and two instances with different labels as a negative sample pair. Thereby, different images with the same semantics are better aggregated in the representation space, while negative sample pairs are separated in the representation space. This makes better use of class-level discriminative information and enhances the generality of the algorithm to some extent.

Finally, our experimental results show that the two-stage framework performs well on cytopathology image with different staining styles. The framework can accurately present negative and positive cell clumps to cytopathologists, providing objective decision support.

The main contributions of our study are as follows:

1. Computer-aided diagnostic studies for endometrial cancer screening are scarce and there is a lack of available datasets. Therefore, our team created an endometrial cancer cytology dataset, which was annotated by two cytopathologists. This dataset contains segmentation labels and classification labels that can be used for deep learning.

2. Compared to histopathology image segmentation, endometrial cytology images have more noise and sparser semantic features, which pose a challenge to segmentation algorithms. We propose a segmentation model based on the UNet architecture, and for better extraction of semantic features in cytology images, we introduce the CA module and the MSS module to learn more local and global contextual representations.

3. Considering that images with the same classification label may be represented differently from each other, e.g., variations in staining styles, which may affect the classification model performance. In order to make full use of the image content information and label information, we propose ECRNet and introduce the label memory bank to make ECRNet focus more on the class-level discriminative information.

4. The framework performs efficiently on H&E-stained and papanicolaou-stained cytopathology images, and shows cytopathologists the negative and positive cell clumps. On the test set, it achieved an average accuracy of 98.50%, an area under the curve (AUC) of 93.66% compared to other classical models. The results show that our model can contribute to medical decision-making.

The rest of this paper is organized as follows: Section 2 describes the materials and methods; Section 3 analyzes our results; Section 4 and Section 5 discuss and conclude our work, respectively.

## Methods

### Data collection

From July 2015, we collected endometrial cells using a sampling device of our own design, the Li Brush (20152660054, Xi'an Meijia Medical Technology Co., Ltd., Xi'an, China). This hospital routine work lasted for seven years since 2015 (XJTU1AHCR2014-007). Until after 2019, endometrial cells obtained from the Li Brush were used in this study (XJTU1AFCRC2019SJ-002). It is important to note that our team spent seven years collecting endometrial cells. However, all data that was used for analysis was obtained after 2019. Therefore, no retrospective ethical approval was involved. The endometrial cells collected from 03/12/2019 to 03/12/2020 used in this study were done so under IRB approval.

From 2019 to 2020, our team collecting images. After 2020 and up to 2022, we are mainly working on the annotation process, and building the endometrial cytopathology image dataset. 139 women who underwent curettage or hysterectomy at the First Affiliated Hospital of Xi'an Jiaotong University were registered in the Obstetrics and Gynecology Registry. Patient exclusion criteria were as follows: (1) diagnosed with suspected pregnancy or pregnancy; (2) diagnosed with acute inflammation of the reproductive system; (3) patients who had undergone hysterectomy for a previous diagnosis of cervical cancer, cervical intraepithelial neoplasia, or ovarian cancer and so on; (4) diagnosed with dysfunctional clotting diseases; and (5) women who body temperature at 37.5°C or higher twice a day were also excluded.

The study is approved by the Ethics Committee of the First Affiliated Hospital of Xi'an Jiaotong University (XJTU1AFCRC2019SJ-002), and written consent was obtained from all patients. Minors were not included in the study. And the authors will not have access to information that could identify individual participants. After three years of collection, 139 patients are eventually included in the study and their details are shown in Table 1, which includes the age of the patients, childbirth history, menstrual status, and any other diseases. The detailed information of age distribution is shown in Fig 1. In addition, histopathological diagnosis was also collected and the specific information is shown in Table 2. It is worth mentioning that the protocols used in this study were all in accordance with the ethical principles of the Declaration of Helsinki on Medical Research [43].

Our study was based on all the cases collected from 2019 to 2020. The data in this work was cleaned so that it did not contain private patient information. The datasets used or analysed during the current study are available from the corresponding author on reasonable request.

We collected endometrial cells with Li Brush, and H&E staining or papanicolaou staining was used for liquid-based cytology specimens of endometrial cells. Histopathological diagnosis of the same patient was also collected at the same time. When the cytopathologic diagnosis was consistent with the histopathologic diagnosis, the case was included in the study. Finally, 39 whole slide images (WSIs) are papanicolaou stained and 100 WSIs are H&E stained.

The MOTIC digital biopsy scanner (EasyScan 60, 20192220065, China) with an ×20 lens was used to scan cytopathological slides in the counterclockwise spiral. And the focal length is automatically adjusted. Since each WSI is very large (e.g., 95200 × 87000 pixels), which cannot be directly input to the deep learning model. We crop the WSIs into same-sized images (1024 × 1024 pixels). Then a simple but effective thresholding algorithm is used to remove the meaningless background images. Specifically, we first calculate the mean and standard deviation

**Table 1. Patient characteristics.**

| Characteristics | Number |
|---|---|
| **SOURCE** | |
| IPD | 81 |
| OP | 58 |
| **AGE** | |
| <40 years old | 32 |
| ≥40 years old | 107 |
| **MENSTRUAL STATUS** | |
| Premenopausal | 77 |
| Postmenopausal | 35 |
| AUB | 27 |
| **OTHER DISEASE** | |
| Ovarian cancer | 18 |
| Hypertension | 24 |
| Diabetes | 24 |
| Hormone replacement therapy | 32 |
| **CHILDBIRTH EXPERIENCE** | |
| Yes | 101 |
| No | 28 |

IPD, Inpatient Department. OP, Outpatient. AUB, Abnormal uterus bleeding. Some information of the patients is missing.

of each image in RGB space. And then, those images with mean between 50 and 230 and standard deviation above 20 are retained. These images often contain meaningful cell clumps.

And the image annotation process includes segmentation label annotation and classification label annotation. Segmentation labels were obtained by two experienced pathologists using Adobe Photoshop CC (2019 v20.0.2.30). First, one senior cytopathologist segmented the cell clumps and the results were reviewed by the other cytopathologist. After the review is accurate, the pathologists annotated the cell clumps according to the International Society of Gynecologic Pathologists and the 2014 World Health Organization classification of uterine tumors. All cell clumps were classified into two categories: malignant (atypical cells of undetermined significance, suspected malignant tumor cells, and malignant tumor cells), and benign (non-malignant tumor cells). Benign diagnosis is defined as cell clumps with neat edges, nuclei with oval or spindle shape, and evenly distributed, finely granular chromatin. Malignant diagnosis referred to a three-dimensional appearance, irregular (including dilated, branched, protruding, and papillotubular) edge, with the nucleus poloidal disordering or disappearing (including megakaryocyte appearance, nuclear membrane thickness, and coarse granular or coarse block chromatin). Both benign and malignant tumors were followed up histologically. Undoubtedly, the cell clumps in the negative slides are all negative, but the ones in the positive slides have both negative and positive cell clumps. Therefore, the two cytopathologists again vote on the labeling of each cell clump, when the votes do not agree, they will discuss it. If the discussion fails to result in an accurate diagnosis, the cell clump is discarded. These measures ensure the accuracy and consistency of segmentation and classification labels.

Based on the results annotated by pathologists, we established the XJTU-EC dataset, which containing 3,620 positive images (endometrial cancer cell clumps and endometrial atypical hyperplasia cell clumps) and 2,380 negative images.

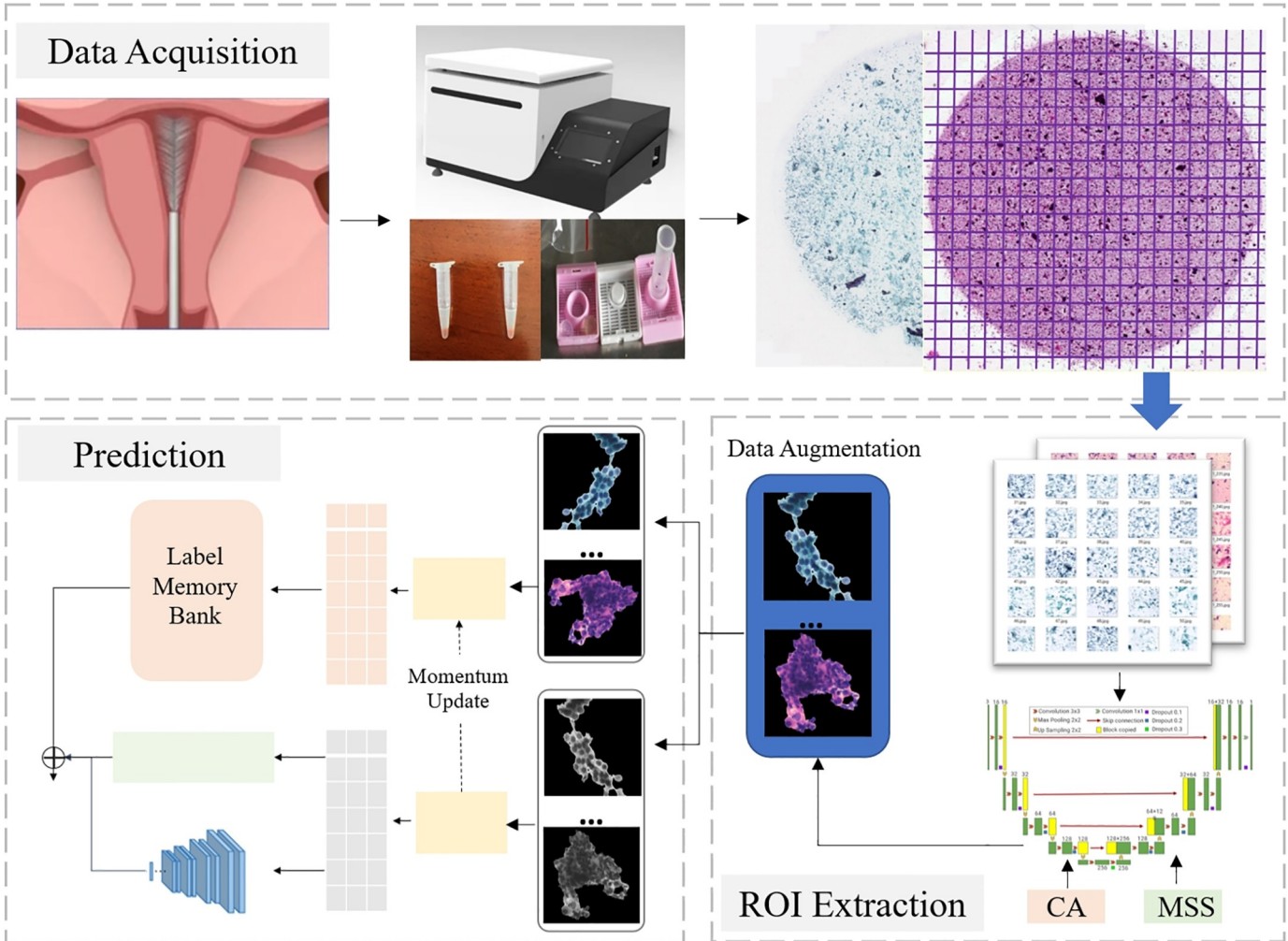

**Fig 1. The age distribution of the patients.**

**Table 2. Pathological diagnosis.**

| Histological diagnostic results | Number |
|---|---|
| Proliferative endometrium | 14 |
| Secretory endometrium | 8 |
| Atrophic endometrium | 10 |
| Mixed endometrium | 2 |
| Endometrial hyperplasia without atypia | 39 |
| Endometrial atypical hyperplasia | 4 |
| **ENDOMETRIAL CARCINOMA** | 62 |
| Endometrioid carcinoma, G1/G2 | 47 |
| Endometrioid carcinoma, G3 | 11 |
| Serous carcinoma | 2 |
| Clear cell carcinoma | 2 |

G1, G2, G3 represent grade 1, grade 2, grade 3 respectively.

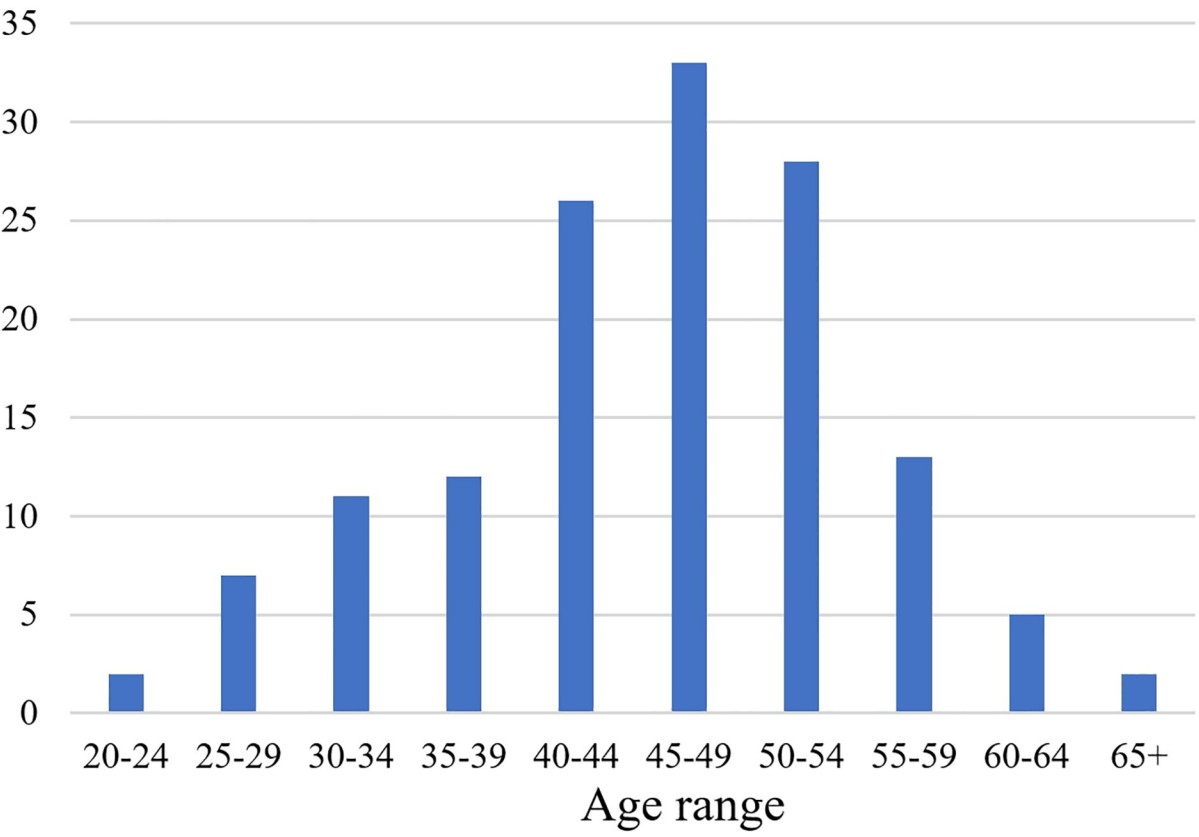

**Fig 2. The proposed pipeline for cancer screening using endometrial cytology data.**

### Endometrial cancer screening

In this paper, we propose a novel framework for endometrial cytology image analysis, applying two fully convolutional networks for early diagnosis. Firstly, a dense connection-based semantic segmentation network, with CA and MSS modules, is used for extract ROIs; secondly, a model based on contrastive learning is applied to classify ROIs. The final results confirm the effectiveness of this strategy. All details as shown in Fig 2.

**CM-UNet.**   To eliminate the interference of neutrophils, dead cells and other impurities contained in the background, and helping the pathologist to better localize the lesion, the first step is to segment the endometrial cell clumps.

The CM-UNet performed ResNet101 as the backbone, and computes the aggregation of all feature maps at each node by applying dense connections [44–46]. As shown in Fig 3.

Let $x^{i,j}$ denote the output of node $X^{I,J}$, then $x^{i,j}$ can be represented as follows:

$$x^{i,j} = \begin{cases} H(D(x^{i-1,j})), & j = 0 \\ H\left(\left[[x^{i,k}]_{k=0}^{j-1}, U(x^{i+1,j-1})\right]\right), & j > 0 \end{cases} \tag{1}$$

where $H(\cdot)$ is a convolution operation followed by an activation function, $D(\cdot)$ and $U(\cdot)$ denote

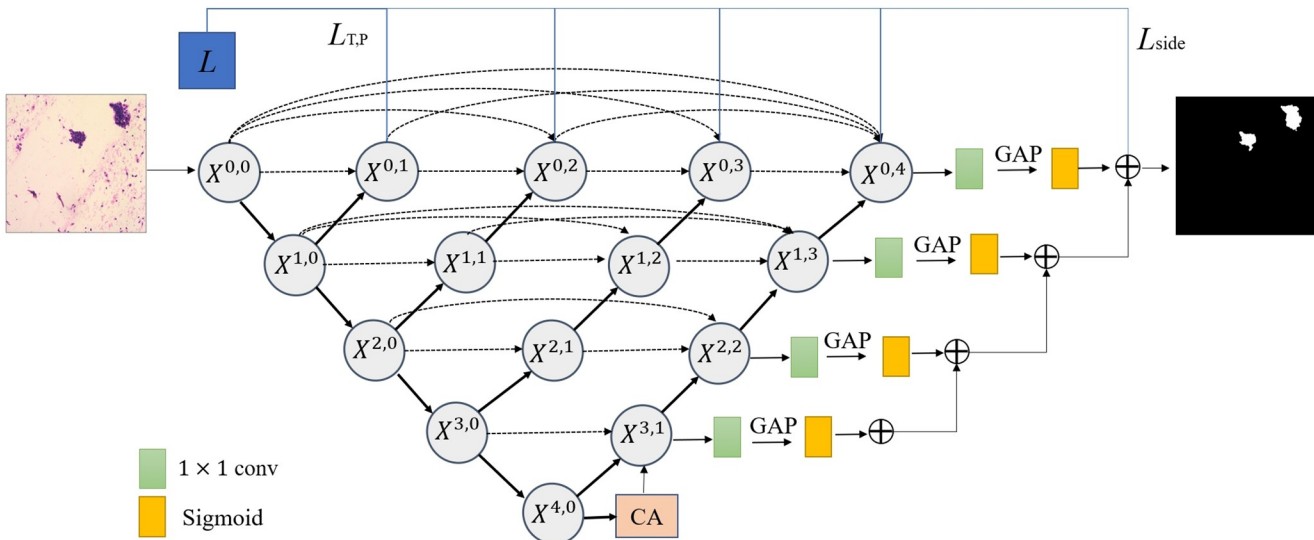

**Fig 3. Pipeline of the proposed framework for cell clumps segmentation.** CM-UNet allows more flexible feature fusion at decoder nodes through densely connected skip connections. $L$ is the loss function. The bold links represent the necessary depth supervision and the light coloured links represent the optional ones.

a down-sampling layer and an up-sampling layer respectively, and [·] denotes the concatenation layer.

To better accommodate different image staining styles, inspired by [47], we apply CA module at the bottleneck of the encoder-decoder network. The CA module integrates the semantic relationships between different channel mappings, and emphasises strongly interdependent channel mappings by adjusting the weights [48]. As shown in Fig 4.

We perform matrix multiplication between the feature maps $x^{4,0}$ and the transpose of $x^{4,0}$. Then apply a softmax layer to calculate the influence of the $a^{th}$ channel on the $b^{th}$ channel:

$$M_{ba} = \frac{\exp(x_a^{4,0} \cdot x_b^{4,0})}{\sum_{a=1}^{K} \exp(x_b^{4,0} \cdot x_b^{4,0})} \quad (2)$$

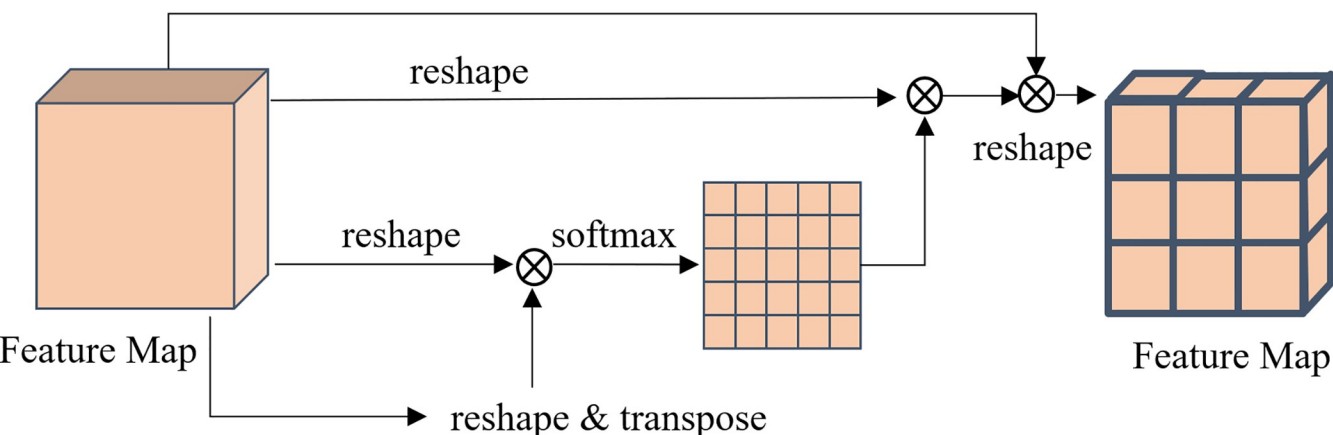

**Fig 4. The details of the channel attention module.**

where *K* is the number of channels. On this basis, the final output *E* is described as follows:

$$E_b = \beta \sum_{a=1}^{K} (M_{ba} x_a^{4,0}) + x_b^{4,0} \tag{3}$$

the scaling parameter $\beta$ is gradually learns a weight from 0, which is updated in subsequent learning.

To solve the gradient disappearance/explosion problem, we introduce the MSS module [49]. By setting appropriate weights for different side output layers, a deeper semantic representation is learned. Assuming that the original input image is represented as *R*, and *d* represents the depth of the CM-UNet model, each output layer *S* performs a $1 \times 1$ convolution operation, followed by the global average pooling to extract global contextual information. And then we assign a weight factor $\alpha$ for each layer of the model. So, the side loss can be defined as:

$$L_{side} = \sum_{i=1}^{d} \alpha_i S_i(R, \theta_i) \tag{4}$$

where $\theta_i$ denotes the relevant parameter of the $i^{th}$ output layer. And $\alpha_3, \alpha_2, \alpha_1, \alpha_0$ is set sequentially to 0.1, 0.3, 0.6, 0.9.

In addition, we introduce a hybrid segmentation loss $L_{T,P}$ to address the class imbalance in the segmentation task:

$$L_{T,P} = -\frac{1}{N} \sum_{c=1}^{C} \sum_{n=1}^{N} (t_{n,c} \log p_{n,c} + \frac{2 t_{n,c} p_{n,c}}{t_{n,c}^2 + p_{n,c}^2}) \tag{5}$$

where $t_{n,c} \in T$, and $p_{n,c} \in P$ denote the ground truth and predicted label for class *c* and $n^{th}$ pixel in the batch. *T* represents the ground truth, and *P* represents the prediction probability. *C* represents the number of categories, and *N* represents the number of pixels in one batch.

Ultimately, the overall loss function of CM-UNet is defined as the weighted sum of the hybrid segmentation loss $L_{T,P}$ and the side loss $L_{side}$. The final loss function is shown as below:

$$L = \sum_{i=1}^{d} (L_{T,P} + L_{side}) \tag{6}$$

where *d* represents the depth of the CM-UNet model.

We trained the segmentation network using the dataset annotated with pathologists, and performed ten-fold cross-validation. It should be noted that segmentation results often have some holes and flaws. Gaps and noise in the images are eliminated by morphological processing. Finally, we obtained a ROI dataset consisting of cell clumps for the next stage.

**ECRNet.** After segmentation of the cytopathological images, noise such as single cells and leukocytes are removed from the background, leaving only the ROIs, as shown in Fig 2. Next, we filled its surroundings with pixels of value 0 until the size was adjusted to $512 \times 512$, in order to further the image analysis task.

Due to the complexity of the endometrial cell features, there is an urgent need for a powerful deep learning classifier to learn and classify cell features. We propose a state-of-the-art method based on contrastive learning to address the above needs, and named ECRNet. The details are shown in Fig 5.

ECRNet consists of two parts: contrastive learning and supervised learning. In contrastive learning, we want to import classification labels in the training data to improve the performance of the classification task. Therefore, we introduce the label memory bank [50]. Two instances with the same label are considered as the same pair, while two instances with different labels are considered as different pairs. This process can be considered as a dictionary

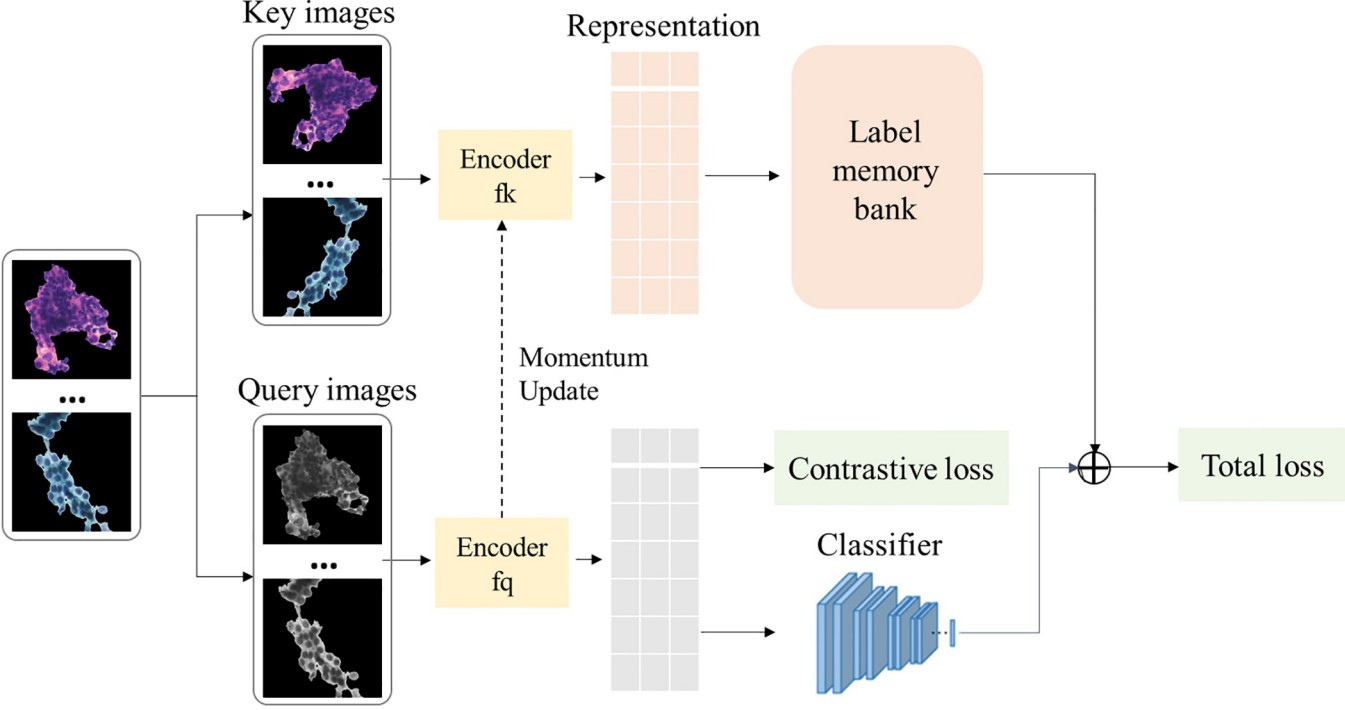

**Fig 5. The details of ECRNet.**

look-up task. Given an encoding query $q$ (with label $y$), we need to look up the corresponding positive key $k$ from a dictionary. Assume that a dictionary of $n$ encoded labeled keys $\{(k_1, y_1), (k_2, y_2),\ldots, (k_n, y_n)\}$, for the given encoded query $(q, y)$, its label contrastive loss $L_{con}$ can be calculated as:

$$L_{con} = -\log \frac{\sum_{i=1}^{n} \mathrm{II}_{y_i=y} \exp(sim(q, k_i)/\tau)}{\sum_{i=1}^{n} \exp(sim(q, k_i)/\tau)} \tag{7}$$

where II is an indicator function that takes the value of 1 if $y$ (the label of the $q$) and $y_i$ (the label of the $k_i$) are the same, otherwise it is 0. $sim(\cdot)$ is a similarity function, and $\tau$ is a temperature parameter.

In order to store the large number of image representations and labels in the label memory bank, we introduce the momentum update method so as to dynamically construct a large and consistent dictionary [51]. This not only reduces the computational overhead, but also allows the learned representations transfer well to the downstream task. The specific update equation is as follows:

$$\theta_k \leftarrow m\theta_k + (1-m)\theta_q \tag{8}$$

Here $m\in$ is a momentum coefficient, taking values between [0,1]. $\theta_k$ is the parameter of the encoder $f_k$, $\theta_q$ is the parameter of the encoder $f_q$.

In supervised learning, we choose the VGG-16 as the classifier [52, 53]. The cross-entropy loss function is our classifier loss functions and defined as follows:

$$L_{cla} = -\frac{1}{|Q|} \sum_{q_i \in Q} \sum_{j \in Y} \mathrm{II}_{q_i,j} \log(p_{q_i,j}) \tag{9}$$

where $Q$ is the set of the query image representations, $Y$ is the label set, $p_{q_i,j}$ is the predicted probability that the query image $q_i$ is predicted to be $j$. II is an indicator function that takes the value of 1 if the query image $q_i$ is classified correctly, otherwise it is 0.

Finally, the ECRNet total loss is calculated as follows:

$$L_{total} = L_{cla} + \beta L_{con} \tag{10}$$

where $\beta$ is a hyperparameter to adjust the relative weight between classification loss and contrastive loss. The value of $\beta$ in general is 0.5.

## Evaluation metrics

We chose the Dice coefficient to evaluate the segmentation model. It is a measure of the similarity between two samples and is one of the commonly used evaluation criteria for segmentation [54]. When the Dice coefficient is 1, it means that the segmentation model achieves perfect results. The Dice coefficient is then calculated as follows:

$$Dice = \frac{2|T \cap P|}{|T| + |P|} \tag{11}$$

where $T$ is the set of ground truth, and $P$ is the set of corresponding segmentation results, respectively.

To better evaluate the performance of the classification model, we use four commonly used quantitative indicators of accuracy, sensitivity, specificity, and F1-Score as the evaluation indicators of the classification model. The indicators are defined as follows:

$$Accuracy = \frac{TP + TN}{TP + FP + TN + FN} \tag{12}$$

$$Sensitivity = \frac{TP}{TP + FN} \tag{13}$$

$$Precision = \frac{TP}{TP + FP} \tag{14}$$

$$F1\_Score = \frac{2(Precision \times Sensitivity)}{Precision + Sensitivity} \tag{15}$$

where $TP$, $TN$, $FP$ and $FN$ represent true positives (correctly classified as positive), true negatives (correctly classified as negative), false positives (incorrectly classified as positive) and false negatives (incorrectly classified as negative), respectively.

In addition, in order to compare the different performance of various classifiers, we select ROC curve and AUC value to visualize the classification results of each classifier. The ROC curve graph reflects the relationship between sensitivity and specificity. Its abscissa represents FPR (false positive rate), and the ordinate is called TPR (true positive rate). The AUC value can be obtained by calculating the area under the ROC curve. A higher AUC value can prove the superiority of the classification model.

## Experiments and results

### Implementation details

The cytopathology images are all augmented by vertical flipping, horizontal flipping, random rotation (90˚, 180˚, 270˚), scaling and graying to improve the framework performance. We use the ImageNet 25 pre-trained weights as the encoder weights to initialize the segmentation and classification models, respectively, while the weights for the decoder part are randomly initialized. And the Adam optimizer is introduced to optimise the model, with an initial learning rate of $5 \times 10^{-3}$ [55]. The temperature parameter $\tau$ is 0.07, the momentum parameter $m$ is 0.9. Finally, the training batch size is set to 32.

We used ten-fold cross-validation to test our models. All networks are implemented based on the TensorFlow framework, and trained by two GPU cards (NVIDIA GeForce GTX 1080), with Python 3.6.12(Python Software Foundation, Wilmington, DE, USA), keras 2.4.3 (Google Brain, Mountain View, CA, USA) and TensorFlow 2.2.0 (Google Brain, Mountain View, CA, USA).

### Segmentation results

We applied our segmentation algorithm and other classical segmentation algorithms, such as fully convolutional networks (FCN) [56], UNet [44], UNet++ [45], LinkNet [57], DeepLabV3 [58], and DeepLabV3+ [59] on HE-stained images and papanicolaou-stained images, respectively. The experimental results are shown in Table 3. Our model shows great potential in segmenting cell clumps, with the average Dice value exceeding 0.85. In addition, we conducted ablation experiments, as shown in Table 3, to verify the role played by the CA module and the MSS module in the segmentation model.

The segmentation results are shown in Fig 6. The first column is the cytological image, the second column is the result of FCN, the third column is the result of UNet, the fourth column is the result of UNet++, the fifth column is the result of LinkNet, the sixth column is the result of DeepLabV3, the seventh column is the result of DeepLabV3+, the eighth column is the result of CM-UNet, and the ninth column is the ground truth annotated by the pathologist. The red boxes represent the over-segmented area, and the yellow boxes represent the under-segmented area. As can be seen from this figure, FCN and UNet have more under-segmentation and fail to identify all cell clumps, which is not suitable for cytopathology image

**Table 3. Comparison experiments.**

| Model | Dice | Training time | Inference time | Params |
|---|---|---|---|---|
| FCN | 0.61 | 2.17h | 0.045s | 270M |
| UNet | 0.75 | 2.50h | **0.022s** | 33M |
| UNet++ | 0.85 | 1.49h | 0.029s | 30M |
| LinkNet | 0.79 | **1.21h** | 0.030s | **12M** |
| DeepLabV3 | 0.81 | 2.30h | 0.030s | 54M |
| DeepLabV3+ | 0.85 | 1.80h | 0.025s | 41M |
| UNet++ (with CA) | 0.86 | 1.52h | 0.030s | 31M |
| UNet++ (with MSS) | 0.88 | 1.88h | 0.039s | 33M |
| CM-UNet | **0.89** | 1.90h | 0.039s | 33M |

Comparison experiment of our segmentation algorithm with other classical segmentation algorithms. The inference time is calculated by the single image.

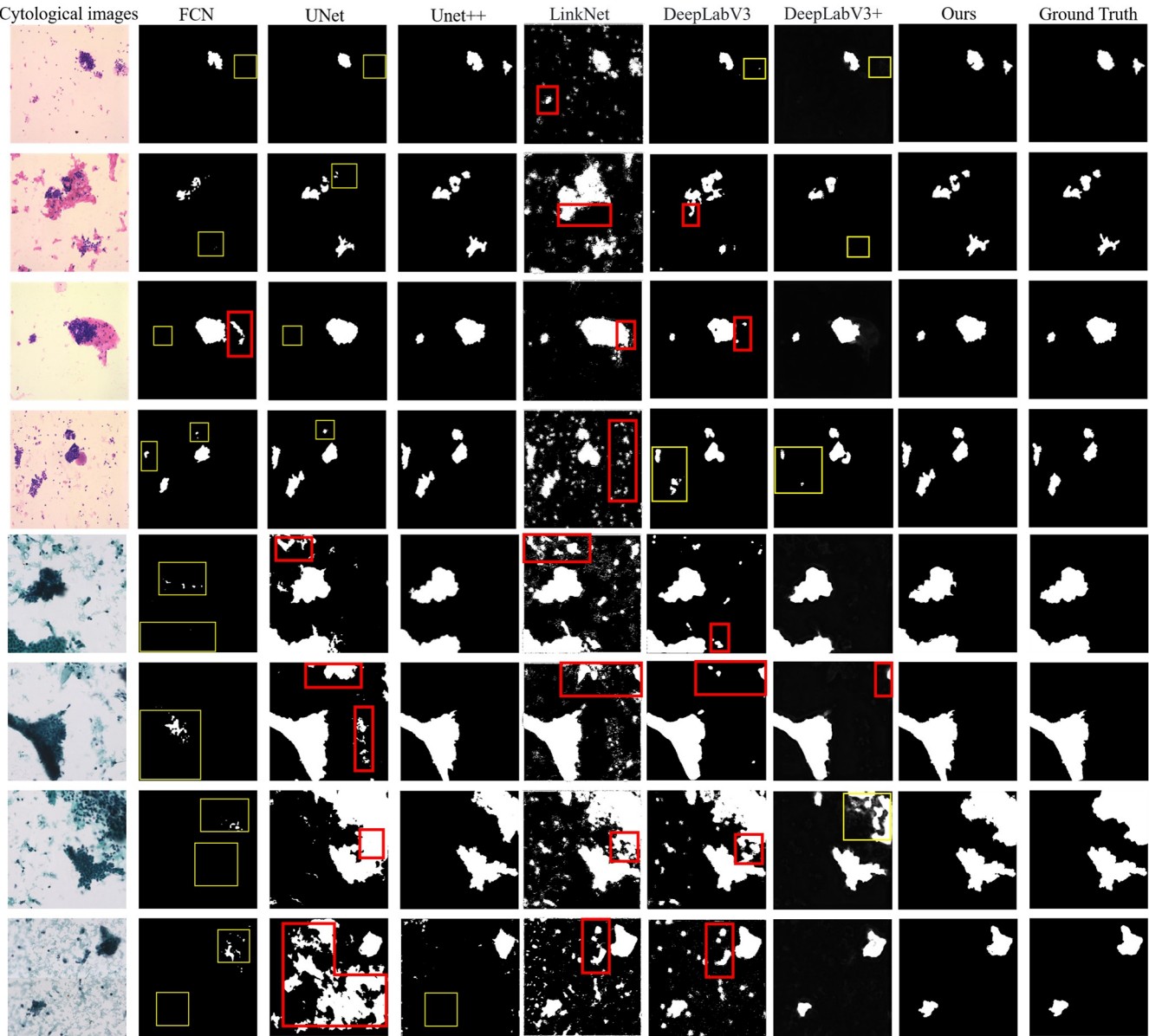

**Fig 6. Comparison with classical segmentation algorithms.** We randomly show the segmentation results of four H&E-stained and four papanicolaou-stained cytopathology images. The red boxes represent the over-segmented area, and the yellow boxes represent the under-segmented area.

segmentation. Whereas LinkNet and DeepLabV3 tend to over-segment, mistaking mucus and single cells for cell clumps, which does not benefit the subsequent classification task and is therefore also not applicable. The segmentation results of UNet++, DeepLabV3+ and CM-U-Net basically conform to the gold standard. However, UNet++ performed moderately well on H&E-stained images but poorly on papanicolaou-stained images, occasionally missing cell clumps. DeepLabV3+, on the other hand, made fewer errors on the papanicolaou-stained images but missed cell clumps on the H&E-stained images. Taken together, the segmentation result of CM-UNet is closer to the annotation of pathologists. It is able to segment all cell clumps and extract ROIs. This demonstrates the performance of our segmentation network.

**Table 4. Ablation experiments.**

| Model | Accuracy (%) | Precision (%) | Recall (%) | F1-score (%) |
|---|---|---|---|---|
| VGG-16 (One stage) | 84.29 | 83.23 | 88.74 | 85.90 |
| VGG-16 (Two stage) | 91.07 | 90.38 | 93.38 | 91.86 |
| ECRNet (One stage) | 89.17 | 88.03 | 90.67 | 89.33 |
| ECRNet (Two stage) | **98.50** | **99.32** | **97.67** | **99.33** |

Table 4 also shows the importance of the segmentation strategy. In the two-stage strategy, we first segment the cytopathology images to obtain ROIs, and apply the classifier to classify the ROIs. In the one-stage strategy, the classifier directly classifies the cytopathology images containing the background. All data sizes are scaled to 512 × 512. The results of the experiment. Among them, VGG-16 performed the worst under the one-stage strategy, with an accuracy of 84.29%. In contrast, ECRNet performed best under the two-stage strategy.

## Classification results

We input the ROIs into ECRNet for ten-fold cross-validation. Table 4 shows the results of the ablation experiments. The backbone (VGG-16) classified the extracted ROI dataset with an accuracy of 91.07%. In contrast, VGG-16 with contrastive learning component achieves 7.43% higher accuracy than backbone.

As shown in Table 5, we compared ECRNet with five classical deep learning models. There are the MobileNet [60], the ResNet-101 [61], the Inception-V3 [62], the ViT [63], the ResNeXt-101 [64], the EfficientNet [65], the DenseNet-121 [66], and VGG-16 [67]. Note that all network parameters remain the same as previously described, and the initialization weights

**Table 5. Comparison with baseline methods.**

| Model | Accuracy (%) | Precision (%) | Recall (%) | F1-score (%) | Params |
|---|---|---|---|---|---|
| VGG-16 +SVM | 78.83 | 75.62 | 78.15 | 76.86 | 138M |
| ResNet-101 +SVM | 88.55 | 84.30 | 85.71 | 85.00 | 24M |
| Inception-V3 +SVM | 87.60 | 80.77 | 88.24 | 84.34 | 22M |
| MobileNet-V1 | 82.99 | 79.11 | 74.79 | 76.89 | **5M** |
| Inception-V3 | 82.17 | 81.43 | 83.33 | 82.37 | 22M |
| ViT | 65.00 | 62.43 | 95.58 | 75.52 | 343M |
| ResNeXt-101 | 86.50 | 81.16 | **99.12** | 89.24 | 79M |
| EfficientNet-B7 | 88.50 | 88.79 | 91.15 | 89.96 | 66M |
| ResNet-101 | 92.17 | 97.03 | 87.00 | 91.74 | 24M |
| DenseNet-121 | 93.50 | 92.23 | 95.00 | 93.59 | 8M |
| Ours | **98.50** | **99.32** | 97.67 | **99.33** | 138M |

In this part, we found that the ViT model performs poorly, which may be due to the small size of our dataset and overfitting of the ViT model. In addition, the MobileNet-V1 model also performs poorly, which may be due to the fact that lightweight networks are not good at learning complex cytopathological features. In contrast, the Inception-V3, the ResNeXt-101, the EfficientNet-B7, the ResNet-101, the DenseNet-121, and the ECRNet models performed better, with mean values of accuracy, precision, recall, and F1-scores above 80%. Specifically, ResNeXt-101 has the highest recall, but has 12% less classification accuracy than ECRNet. And ResNet-101 has a precision of 97.03%, second only to ECRNet, which indicates that it has a lower false positive rate. However, the recall of ResNet-101 is only 87.00%, which indicates that it has a higher false negative rate. And it is more likely to miss cancer cell clumps. And ECRNet has the best classification performance, outperforming other classification models in terms of accuracy, precision and F1-score.

are the ImageNet pre-trained weights. These results are obtained in the two-stage framework, which is based on the classification of cell clumps.

Furthermore, several studies [68] show that using CNNs to extract features and train linear support vector machines (SVMs) achieves better performance than end-to-end CNN-based classifiers. Therefore, we use three classical CNNs, that is VGG-16, ResNet-101 and Inception-V3, to obtain the image feature vectors. These feature vectors are then used to train a linear SVM that classifies all ROIs as positive or negative. As the dimensionality of the feature maps is large, we used principal component analysis to reduce the dimensionality of the image features. The SVM classifier uses a radial basis function kernel with parameters γ and C set to 0.0078 and 2, respectively. And the rest of the experimental settings are consistent with those described previously. The results are shown in Table 5. As can be seen, all results are unsatisfactory, with VGG-16+SVM having an accuracy of only 78.83%.

We further plotted the ROC curves for the classifiers in the binary classification task. As can be seen from Fig 7, ECRNet outperformed all the models. This experiment suggests that the model benefits more from two-stage framework than one-stage framework. Our two stage strategy is effective.

Finally, we also discuss the ECRNet classification failure cases. As shown in Fig 8, we randomly listed 8 correctly classified cell clumps and 8 failure cases in the test set. Of these, the 4 false-negative (missed diagnosis) cases consisted of 1 well-differentiated endometrial adenocarcinoma and 3 poorly differentiated endometrial adenocarcinomas. In contrast, 4 false-positive (over diagnosed) cases included 3 normal cell clumps that were classified as cancer. In some of these classification failure samples, it was difficult for the classification model to extract deeper features because of cell stacking and obscure structural features. In addition, another part of the failed cases showed that the number of cells in the image was small, which was easier to be misclassified by the classification model. This suggests that ECRNet's ability to classify small targets needs to be strengthened in future work.

In addition, we conducted an external validation of the paper's algorithm using the public dataset. The externally validated data were obtained from the public data platform, AIstudio, accessible via the Internet [69]. It is worth noting that the data used for external validation came from the public dataset and did not contain segmentation labels, so this external validation mainly evaluated the classification performance of ECRNet.

The dataset used for external validation consisted of 848 negative endometrial cytopathology images and 785 positive endometrial cytopathology images. The ratio of negative to positive images was 1.08:1. All images are papanicolaou stained images. During this external validation, we also chose four classification models (the ResNet-101 model, the DenseNet-121 model, the EfficientNet-B7 model, and the ResNeXt-101 model) to compare with our model. This is because these four models perform well in the classification task, second only to ECR-Net. Specifically, these models above were first trained using images (1024 × 1024 pixels) from the XJTU-EC dataset instead of cell clumps (ROIs), and then tested directly on an external dataset. The specific results are shown in Table 6 as following:

In this external validation experiment, ResNet-101 has the highest precision of 100%, which means it has no false positives in the external validation. However, both ResNeXt-101, EfficientNet-B7 ResNet-101, and DenseNet-121 have low recall, which means they can easily miss screening positive patients. This is unacceptable for clinical tasks. In contrast, our model achieved the highest recall of 96.17%. In addition, ECRNet has the highest accuracy of 95.32%, followed by the DenseNet classifier with 83.53%. It is worth noting that ECRNet significantly outperforms the other four classifiers in terms of the F1 score. The F1 score is the reconciled average of precision and recall, which combines the information of precision and recall to provide a more comprehensive assessment of the performance of the classifiers, and the higher

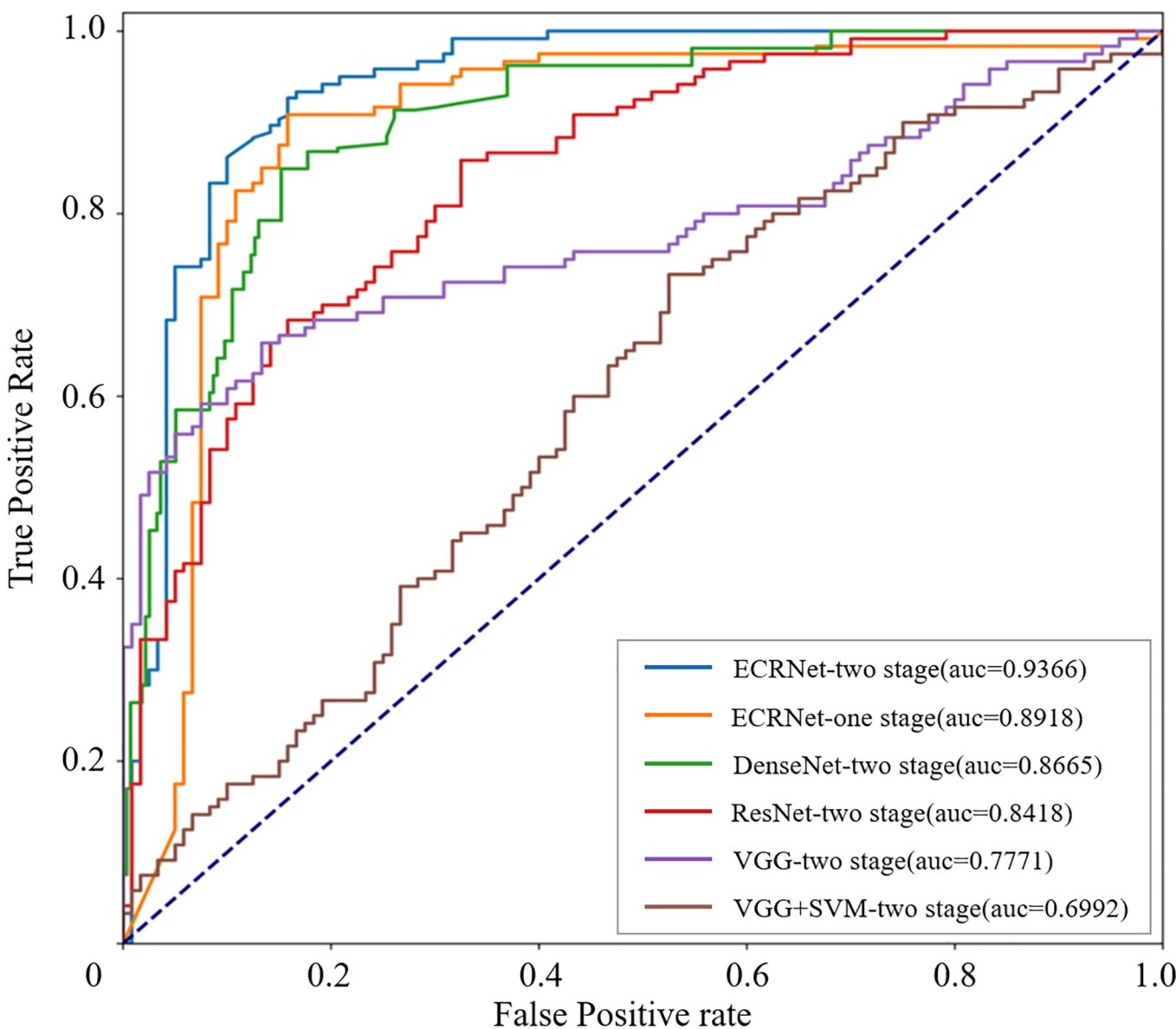

**Fig 7. The false positive rate, the y-axis represents the true positive rate.** The points above the diagonal line are indicated by dashed lines indicating a better than random classification result, i.e. an AUC value of 0.5.

the F1 score, the better the performance of the classifiers. The higher the F1 score, the better the performance of the classifier. In summary, ECRNet has the best performance in this external validation experiment.

## Discussion

Currently, there is no well-established method to screen endometrial cancer. The main screening tests for endometrial cancer include ultrasound, hysteroscopy and endometrial biopsy. Sequential transvaginal ultrasound scan is a less invasive method of assessment, but lacks a high degree of specificity. Until now, the collection of tissue samples from the endometrium and analysis of histopathological images by physicians has been the gold standard for the

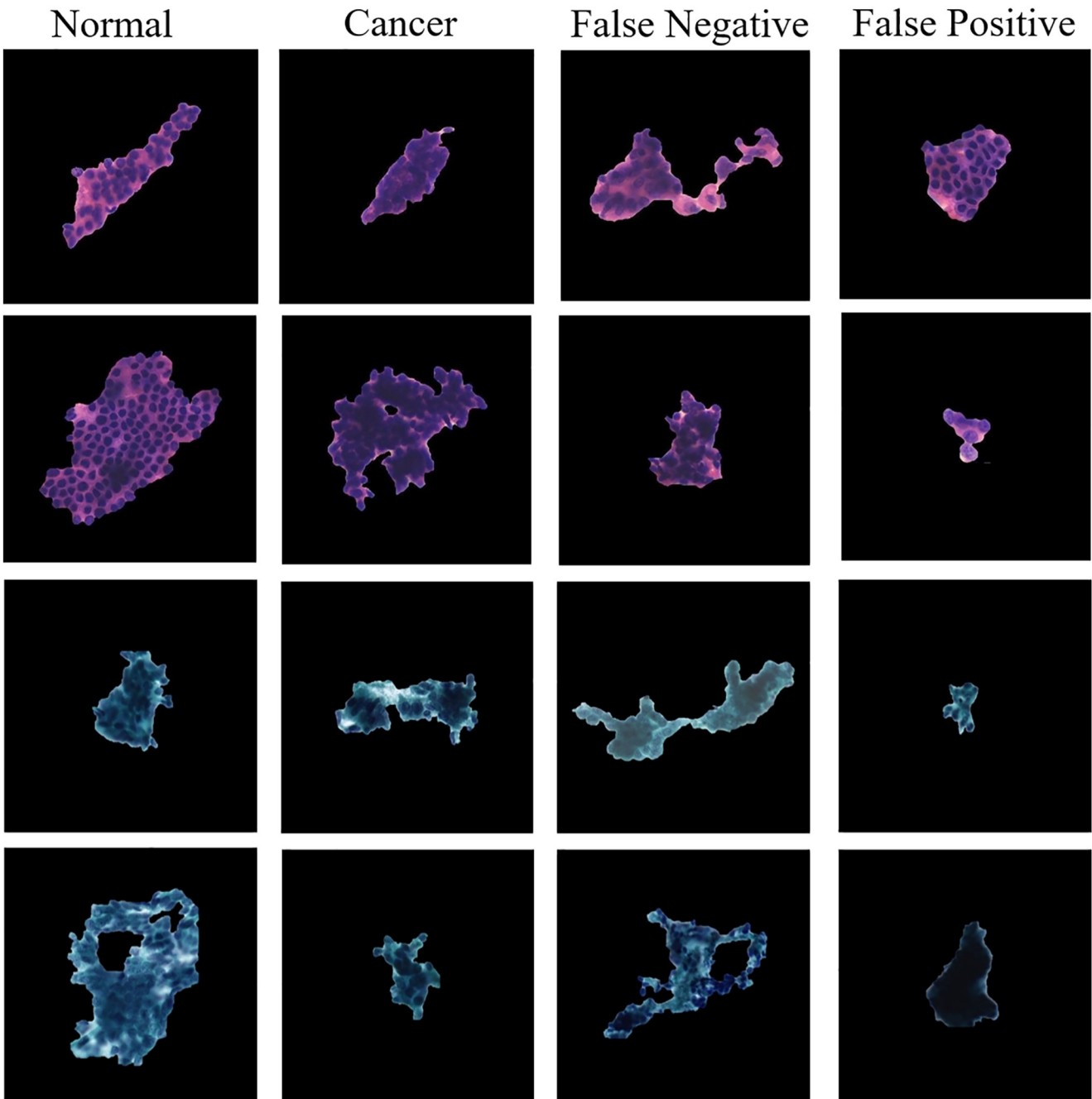

**Fig 8. Examples of correct and incorrect predictions of ECRNet.**

diagnosis of endometrial cancer. However, both endometrial biopsy and hysteroscopy are invasive and require the cooperation of anaesthetists, which is expensive. As a result, cytopathology-based screening for endometrial cancer is becoming increasingly desirable.

Due to the lack of relevant data and the complexity of cell morphology, endometrial cancer screening based on cytopathology is difficult to promote. Therefore, our team spent three years collecting and annotating WSIs from 139 patients to create the endometrial

**Table 6. External validation comparison results.**

| Model | Accuracy (%) | Precision (%) | Recall (%) | F1-score (%) |
|---|---|---|---|---|
| ResNeXt-101 | 64.50 | 86.20 | 44.20 | 58.44 |
| EfficientNet-B7 | 83.53 | 78.72 | 59.2 | 67.58 |
| ResNet-101 | 73.50 | **100.00** | 53.10 | 69.37 |
| DenseNet-121 | 80.10 | 77.20 | 67.20 | 71.90 |
| Ours | **95.32** | 94.57 | **96.17** | **95.37** |

cytopathology image dataset, named the XJTU-EC dataset. Since our dataset contains both papanicolaou and H&E stained images, which are the most common staining modalities for cytology images. Therefore, it can be somewhat considered a representative dataset. In addition, the data includes patients of different age, so the XJTU-EC dataset has more diversity. Based on this dataset, we investigated the first clinically automated deep learning framework for extracting and identifying normal or cancerous endometrial cell clumps. The results will be presented to cytopathologists as a reference.

In order to adapt to different staining styles, in the cell clump extraction stage, we use the robust UNet as the backbone, which has been previously generalized to many datasets. Based on this, we introduced the CA module pay attention to global contextual information, and MSS module to aggregate semantic features at multiple scales. So, our method achieves better segmentation results. Experiments demonstrate that CM-UNet is able to perform well on both H&E-stained images and papanicolaou-stained images.

In the cell clump classification stage, we design an ECRNet based on contrastive learning, which considers both instances and label facts. Specifically, different staining style images with the same classification label are considered similar. In addition, we learn meaningful and consistent anatomical features through the label contrastive loss, and introduce a label memory bank and a momentum update encoder to maintain encoded feature consistency. Experimental results show that our method achieves excellent performance on mixed staining style datasets, indirectly demonstrating its robustness. Compared to other methods, ECRNet achieves the best performance in both classification tasks with the two-stage strategy and the one-stage strategy.

Finally, there are two limitations of this work. On the one hand, because the data comes from a single institution, our approach is not externally validated on different institutional datasets. Although we have tried to ensure as much diversity as possible in the dataset during the data collection process, and used contrastive learning to enhance the generalization of the screening framework. However, we still lack external validation results from different medical centers. In future work, we will extend our method to other medical center datasets for external validation. In addition, annotation has been a challenge due to the scarce number of cytopathologists. We will focus on investigating self-supervised learning to reduce the annotation workload of cytopathologists.

## Conclusions

In this paper, we present a clinically motivated deep learning framework for endometrial cancer cell clumps screening. in the first stage, we propose CM-UNet to obtain the ROI set, and the CA and MSS modules are able to fuse features from different scales to obtain more semantic information. In the second stage, we utilize ECRNet to classify ROIs. Contrastive learning is used to bring instances of the same class in the representation space closer together and push instances of different classes apart. Experiments show that our framework performs well on the XJTU-EC dataset. Our future work will focus on providing objective and

complementary diagnostic input for clinical diagnosis, and supporting effective deployment by advanced algorithms. We believe that this can help reduce the burden on patients and physicians.

## Author Contributions

**Conceptualization:** Qiling Li.

**Data curation:** Qing Li, Qiling Li.

**Funding acquisition:** Dexing Zhong.

**Investigation:** Guizhi Shi.

**Methodology:** Ruijie Wang.

**Validation:** Guizhi Shi, Qiling Li.

**Visualization:** Guizhi Shi, Qiling Li.

**Writing – original draft:** Ruijie Wang.

**Writing – review & editing:** Ruijie Wang.

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
