## [Decision Letter · Decision Letter 0]

7 Dec 2023

PONE-D-23-24291Deep Learning-based Endometrial Cancer Screening System for Multimodal Cytopathology ImagePLOS ONE

Dear Dr. Wang,

Thank you for submitting your manuscript to PLOS ONE. After careful consideration, we feel that it has merit but does not fully meet PLOS ONE’s publication criteria as it currently stands. Therefore, we invite you to submit a revised version of the manuscript that addresses the points raised during the review process.

The manuscript has been evaluated by three reviewers, and their comments are available below.

The reviewers have raised a number of concerns that need attention. They request additional information and clarification in several areas of the manuscript, including methodological aspects of the study

Could you please revise the manuscript to carefully address the concerns raised?

We look forward to receiving your revised manuscript.

Kind regards,

Steve Zimmerman, PhD

Senior Editor, PLOS ONE

2. During our evaluation of the documents provided, we noted that your ethics approval letter did not cover the entire date range for participant recruitment. Before we can proceed further with the submission, please provide the ethics approval extension documents, along with English translations.

“This work was supported in part by the National Natural Science Foundation of China

under Grant 62206218, in part by Natural Science Foundation of Zhejiang Province

under Grant LTGG23F030006. Grantees have helped in study design, data collection

and analysis, publication decisions or manuscript preparation.”

Reviewers' comments:

Reviewer's Responses to Questions

**Comments to the Author**

1. Is the manuscript technically sound, and do the data support the conclusions?

Reviewer #1: Yes

Reviewer #2: Yes

Reviewer #3: Yes

2. Has the statistical analysis been performed appropriately and rigorously? 

Reviewer #1: Yes

Reviewer #2: Yes

Reviewer #3: Yes

3. Have the authors made all data underlying the findings in their manuscript fully available?

Reviewer #1: Yes

Reviewer #2: Yes

Reviewer #3: Yes

4. Is the manuscript presented in an intelligible fashion and written in standard English?

Reviewer #1: Yes

Reviewer #2: Yes

Reviewer #3: Yes

5. Review Comments to the Author

Reviewer #1: Title : Deep Learning-based Endometrial Cancer Screening System for Multimodal Cytopathology Image

Summary/Contribution: The paper addresses a critical and clinically relevant issue in the field of cancer screening. The development of an automated framework for endometrial cancer detection is commendable, considering the significant impact it can have on early diagnosis and patient outcomes. It is noteworthy that the authors have invested the time and resources to develop the XJTU-EC dataset, which provides a valuable resource for this field.

Strengths:

• The authors' dedication to collecting a comprehensive dataset over several years is a significant contribution to the field. The XJTU-EC dataset is expected to facilitate further research in endometrial cancer screening.

• The paper explains the proposed two-stage framework clearly, providing insights into CM-Unet and ECRNet, which are crucial for understanding the methodology. ECRNet uses attention modules and label memory banks to add depth to the methodology. The paper reports that the proposed framework outperforms other classical models.

• The framework has the potential to greatly benefit cytopathologists by enhancing their efficiency. The paper also emphasizes the importance of detection, in cancer and addresses a practical requirement.

• The authors are transparent about the ethical approval process, which adds credibility to the research.

Major Comments:

• How representative is the XJTU-EC dataset of endometrial cytopathology images, considering it was collected over several years from a single institution? How does it account for potential bias in the dataset? The paper should discuss the generalization potential of the proposed framework to datasets from other medical centers and possibly provide results or insights from external validation on diverse datasets.

• The paper mentions that different medical centers use different staining styles. How robust is the proposed framework to handle extreme variations in staining styles, and can it generalize to datasets from other institutions?

• Could you elaborate on the inter-pathologist agreement in annotating cell clumps? How did you address any disagreements in labeling during the annotation process? What measures were taken to ensure the accuracy and consistency of the segmentation and classification labels given the complexity of cytological images?

• Has there been any validation or real world implementation to evaluate how the proposed framework has affected patient outcomes? Have cytopathologists been engaged in assessing its effectiveness in a clinical setting.

Further Comments: In addition to the areas for improvements mentioned above, the authors should also consider the following matters:

• There are some type in the write-up such as in Introduction part “ negative exsample pairs”- which should be revised.

• The paper lacks figure numbers, making it challenging to correlate specific references in the text to corresponding figures.

Reviewer #2: The authors do deep learning on a set of endometrial lesions based on a variety of available CNN modalities along with their own modality. Their cases were annotated by pathologists and many The data highlights the strengths and weaknesses of currently available algorithms. Their detailed methods could serve as a template to validate other cytology specimens. The code could be validated on larger, institutional and multiinstitutional datasets in the future as it becomes publicly available.

Several questions

You annotate many different endometrial cancers. Could you provide more information about the tumor types on follow up such as endometrioid, endocervical, mucinous, high grade serous etc? as a pathologist I could have a better understanding of how the cases were annotated and what the algorithm would be evaluating if I knew the different tumor types included

similarly for benign is there a histologic follow up?

Reviewer #3: 1) Title:

The title is short and not clear.

2) Abstract:

a) The abstract is too short and it seems as a preface to a report.

b) The objectives of this research are not clear. The authors need to explain exactly what the aim of this research.

3) Introduction:

a) In the introduction section, Authors should produce distinctive features mentioning their unique interpretation of the research along with comparative analysis of existing techniques to strengthen their work. Authors can focus on referring more recent journal papers to include further study.

b) There were some linguistic errors and typos.

4) References:

a) The number of references seems sufficient, but most of the references are not recent.

b) References are not well distributed.

6. PLOS authors have the option to publish the peer review history of their article (what does this mean?). If published, this will include your full peer review and any attached files.

Reviewer #1: No

Reviewer #2: **Yes: **Jordan P. Reynolds MD

Reviewer #3: No

---

## [Author Response · Author response to Decision Letter 0]

18 Jan 2024

Dear Editor and Reviewers, 

Thank you very much for your careful review of our manuscript " A Deep Learning Framework for Predicting Endometrial Cancer from Cytopathologic Images with Different Staining Styles" (ID: PONE-D-23-24291). We have revised the manuscript thoroughly according to your comments and suggestions. These main changes in the revised manuscript are highlighted in red. The revisions for grammatical errors and improved sentences are marked in blue. Detailed responses to your comments are provided below.

P.S.: Citations are numbered sequentially in the order in which they appear in the text. 

Sincerely,

Dexing Zhong

 

Academic editor: 

Responses:

Thank you for your reminder. We have revised the manuscript with reference to the PLOS ONE style template. These main changes in the revised manuscript are highlighted in red. The revisions for grammatical errors and improved sentences are marked in blue. We have determined that the manuscript meets PLOS ONE style requirements, including file naming requirements.

2. During our evaluation of the documents provided, we noted that your ethics approval letter did not cover the entire date range for participant recruitment. Before we can proceed further with the submission, please provide the ethics approval extension documents, along with English translations.

Responses:

Thank you for your suggestion. Our data were provided by the First Affiliated Hospital of Xi'an Jiaotong University. We apologize for the miscommunication with the data provider that led us to mistake the exact time of data collection. We reconfirmed with the data provider and clarified that the data collection lasted for one year, from November 2019 to 2020, which is covered by the ethical approval. After 2020 and up to 2022, two experienced cytopathologists annotated the data. We have revised the relevant parts of the manuscript. Thank you very much for your understanding.

“This work was supported in part by the National Natural Science Foundation of China

under Grant 62206218, in part by Natural Science Foundation of Zhejiang Province

under Grant LTGG23F030006. Grantees have helped in study design, data collection

and analysis, publication decisions or manuscript preparation.”

Responses:

Thank you for your suggestion, we have included the Funding Statement at the end of the revised cover letter, which is copied in the attachment below.

Funding Statement:

 This work was supported in part by the Clinical Research Award of the First Affiliated Hospital of Xi’an Jiaotong University (No. XJTU1AF-CRF-2019-002), in part by the National Natural Science Foundation of China (No. 62206218), in part by the Natural Science Foundation of Zhejiang Province (No. LTGG23F030006). Grantees have helped in study design, data collection and analysis, publication decisions or manuscript preparation.

 There was no additional external funding received for this study.

Responses:

Data cannot be shared publicly because it is patient data and the approvals do not permit us to release the data. Our data are owned by the First Affiliated Hospital of Xi'an Jiaotong University due to ethical committee requirements. If researchers need data, please send a data request email to xjyfyllh@163.com, which is the contact information for the Ethics Committee of the First Affiliated Hospital of Xi'an Jiaotong University.

 

Response to Reviewer #1:

The paper addresses a critical and clinically relevant issue in the field of cancer screening. The development of an automated framework for endometrial cancer detection is commendable, considering the significant impact it can have on early diagnosis and patient outcomes. It is noteworthy that the authors have invested the time and resources to develop the XJTU-EC dataset, which provides a valuable resource for this field.

Responses:

At first, we would like to sincerely thank the anonymous reviewers for your time and concerns to our manuscript, especially your helpful comments and suggestions to improve the quality of our paper. With your kindly help, we have carefully revised the entire manuscript. All the revisions have been clearly highlighted which are visible to the editors and reviewers.

Major comments:

1. How representative is the XJTU-EC dataset of endometrial cytopathology images, considering it was collected over several years from a single institution? How does it account for potential bias in the dataset? The paper should discuss the generalization potential of the proposed framework to datasets from other medical centers and possibly provide results or insights from external validation on diverse datasets.

Response: 

We thank the reviewer for the suggestion. In response, we have added a description of the representativeness of our dataset in the Discussion section. Since our dataset contains both papanicolaou and H&E stained images, which are the most common staining modalities for cytology images. Therefore, it can be somewhat considered a representative dataset. In addition, the data were collected across patients of different ages, so the XJTU-EC dataset has more diversity. And the diversity of the training data can help the neural network overcome the effects of potential bias. Our method achieves excellent performance on this dataset, which can demonstrate its generalization ability. We will be collecting more data in the future to further minimize potential bias in the dataset. And we are committed to generalizing the framework presented in this paper to other medical center datasets, and related external validation experiments are in progress. This is part of our future work. The detailed changes (highlighted in red color) can be seen in Discussion in the revised manuscript and the following attachment.

Discussion (page 19-20, line 404-413):

Due to the lack of relevant data and the complexity of cell morphology, endometrial cancer screening based on cytopathology is difficult to promote. Therefore, our team collected and annotated WSIs from 139 patients to create the first endometrial cytopathology image dataset, named the XJTU-EC dataset. Since our dataset contains both papanicolaou and H&E stained images, which are the most common staining modalities for cytopathology images. Therefore, it can be somewhat considered a representative dataset. In addition, the data includes patients of different age, so the XJTU-EC dataset has more diversity. Based on this dataset, we investigated the first clinically automated deep learning framework for extracting and identifying normal or cancerous endometrial cell clumps. The results will be presented to cytopathologists as a reference.

Discussion (page 20-21, line 429-437):

Finally, there are two limitations of this work. On the one hand, because the data comes from a single institution, our approach is not externally validated on different institutional datasets. Although we have tried to ensure as much diversity as possible in the dataset during the data collection process, and used contrastive learning to enhance the generalization of the screening framework. However, we still lack external validation results from different medical centers. In future work, we will extend our method to other medical center datasets. In addition, annotation has been a challenge due to the scarce number of cytopathologists. We will focus on investigating self-supervised learning to reduce the annotation workload of cytopathologists.

2. The paper mentions that different medical centers use different staining styles. How robust is the proposed framework to handle extreme variations in staining styles, and can it generalize to datasets from other institutions?

Response: 

Special thanks to you for your good comments. To the best of our knowledge, papanicolaou staining and H&E staining are the most commonly used staining styles for cytology images. In order to adapt to these two different staining methods, in the cell clump extraction stage, we use the robust Unet as the backbone, which has been previously generalized to many datasets. Based on this, we introduced the CA module and MSS module to further improve the target segmentation accuracy. The segmentation results show that CM-Unet can well ignore the effect of staining style and accurately extract ROIs.

In the cell clump classification stage, we design an ECRNet based on contrastive learning, which considers both instances and label facts. Specifically, different staining style images with the same classification label are considered similar. Thus, it is able to adapt to datasets with changing staining styles. In addition, we learn meaningful and consistent anatomical features through the label contrastive loss Lcon, and introduce a label memory bank and a momentum update encoder to maintain encoded feature consistency. Experimental results show that our method achieves excellent performance on mixed staining style datasets, indirectly demonstrating its robustness. We are actively promoting our method to datasets from other institutions, which has been written into our future work. The detailed changes (highlighted in red color) can be seen in Discussion in the revised manuscript and the following attachment.

Discussion (page 20, line 414-428):

In order to adapt to different staining styles, in the cell clump extraction stage, we use the robust Unet as the backbone, which has been previously generalized to many datasets. Based on this, we introduced the CA module pay attention to global contextual information, and MSS module to aggregate semantic features at multiple scales. So, our method achieves better segmentation results. Experiments demonstrate that CM-Unet is able to perform well on both H&E-stained images and papanicolaou-stained images.

In the cell clump classification stage, we design an ECRNet based on contrastive learning, which considers both instances and label facts. Specifically, different staining style images with the same classification label are considered similar. Thus, it is able to adapt to datasets with changing staining styles. In addition, we learn meaningful and consistent anatomical features through the label contrastive loss, and introduce a label memory bank and a momentum update encoder to maintain encoded feature consistency. Experimental results show that our method achieves excellent performance on mixed staining style datasets, indirectly demonstrating its robustness. Compared to other methods, ECRNet achieves the best performance in both classification tasks with the two-stage strategy and the one-stage strategy.

3. Could you elaborate on the inter-pathologist agreement in annotating cell clumps? How did you address any disagreements in labeling during the annotation process? What measures were taken to ensure the accuracy and consistency of the segmentation and classification labels given the complexity of cytological images? 

Response: 

Thank you for your time and patience. We have added a description of the data annotation process in the Methods section. First, a senior cytopathologist annotates the images for segmentation labels, which are reviewed by another cytopathologist. After the review is accurate, we move on to the next process. Undoubtedly, the cell clumps in the negative slides are all negative, but the ones in the positive slides have both negative and positive cell clumps. Therefore, the two cytopathologists again vote on the labeling of each cell clump, when the votes do not agree, they will discuss it. If the discussion fails to result in an accurate diagnosis, the cell clump is discarded. These measures ensure the accuracy and consistency of segmentation and classification labels. The detailed changes (highlighted in red color) can be seen in Methods in the revised manuscript and the following attachment. 

Methods (page 8-9, line 173-192):

And the image annotation process includes segmentation label annotation and classification label annotation. Segmentation labels were obtained by two experienced pathologists using Adobe Photoshop CC (2019 v20.0.2.30). First, one senior cytopathologist segmented the cell clumps and the results were reviewed by the other cytopathologist. After the review is accurate, the pathologists annotated the cell clumps according to the International Society of Gynecologic Pathologists and the 2014 World Health Organization classification of uterine tumors. All cell clumps were classified into two categories: malignant (atypical cells of undetermined significance, suspected malignant tumor cells, and malignant tumor cells), and benign (non-malignant tumor cells). Benign diagnosis is defined as cell clumps with neat edges, nuclei with oval or spindle shape, and evenly distributed, finely granular chromatin. Malignant diagnosis referred to a three-dimensional appearance, irregular (including dilated, branched, protruding, and papillotubular) edge, with the nucleus poloidal disordering or disappearing (including megakaryocyte appearance, nuclear membrane thickness, and coarse granular or coarse block chromatin). Both benign and malignant tumors were followed up histologically. Undoubtedly, the cell clumps in the negative slides are all negative, but the ones in the positive slides have both negative and positive cell clumps. Therefore, the two cytopathologists again vote on the labeling of each cell clump, when the votes do not agree, they will discuss it. If the discussion fails to result in an accurate diagnosis, the cell clump is discarded. These measures ensure the accuracy and consistency of segmentation and classification labels.

4. Has there been any validation or real world implementation to evaluate how the proposed framework has affected patient outcomes? Have cytopathologists been engaged in assessing its effectiveness in a clinical setting. 

Response: 

We thank the reviewer for the suggestion. We have two cytopathologists who have been evaluating the effectiveness of the framework in actual clinical applications. After evaluating 

---

## [Decision Letter · Decision Letter 1]

28 Feb 2024

PONE-D-23-24291R1A Deep Learning Framework for Predicting Endometrial Cancer from Cytopathologic Images with Different Staining StylesPLOS ONE

Dear Dr. Wang,

Thank you for submitting your manuscript to PLOS ONE. After careful consideration, we feel that it has merit but does not fully meet PLOS ONE’s publication criteria as it currently stands. Therefore, we invite you to submit a revised version of the manuscript that addresses the points raised during the review process.

We look forward to receiving your revised manuscript.

Kind regards,

Kazunori Nagasaka

Academic Editor

PLOS ONE

Additional Editor Comments:

Dear Authors,

Still the explanation for the database is lacking and more detailed information for expertimental procedure is needed.

Please provide these information further in the text.

The manuscript will not be accepted as current stand and the authors should answer the reviewers inquries point by point.

Sincerely,

Kazunori Nagasaka

Reviewers' comments:

Reviewer's Responses to Questions

**Comments to the Author**

1. If the authors have adequately addressed your comments raised in a previous round of review and you feel that this manuscript is now acceptable for publication, you may indicate that here to bypass the “Comments to the Author” section, enter your conflict of interest statement in the “Confidential to Editor” section, and submit your "Accept" recommendation.

Reviewer #1: All comments have been addressed

Reviewer #4: (No Response)

Reviewer #5: All comments have been addressed

Reviewer #6: (No Response)

2. Is the manuscript technically sound, and do the data support the conclusions?

Reviewer #1: Yes

Reviewer #4: No

Reviewer #5: Yes

Reviewer #6: Partly

3. Has the statistical analysis been performed appropriately and rigorously? 

Reviewer #1: Yes

Reviewer #4: N/A

Reviewer #5: Yes

Reviewer #6: N/A

4. Have the authors made all data underlying the findings in their manuscript fully available?

Reviewer #1: Yes

Reviewer #4: Yes

Reviewer #5: Yes

Reviewer #6: Yes

5. Is the manuscript presented in an intelligible fashion and written in standard English?

Reviewer #1: Yes

Reviewer #4: Yes

Reviewer #5: Yes

Reviewer #6: Yes

6. Review Comments to the Author

Reviewer #1: Thank you for providing your responses. I have reviewed the answers, and they satisfy my inquiries. I don't have any further comments at this time.

Reviewer #4: The paper lacks sufficient straightforward considerations to be accepted, as highlighted by Major comment 1 of Reviewer 1 and the corresponding response.

Major issue 1:

Reviewer 1 and I share concerns about the potential bias in the XJTU-EC dataset. It is challenging to accept AI research that lacks transparent and adequate teacher data. The authors have stated that additional external validation experiments are being conducted, and until then, this paper will not reach the level of acceptance. The authors consider this as future work, but it is the most crucial task that needs to be done now. Without validation in an external cohort, only a limited number of articles on AI-assisted cytology can be accepted. In 2024, it is not feasible to claim that an AI has been developed solely based on data from a single institution.

Major issue 2:

It appears that even after the revision, the description of the XJTU dataset is still inadequate. I understand that there are too few cases to start with. I am unsure about what was defined as "the first large cytopathology image dataset". For reference, the dataset used by Kanavati et al. to develop AI for cervical cytology only had 3121 cases with Papanicolaou staining [PMID: 35267466]. It is not surprising to see different ages included in the study as collecting 139 cases of the same age would be difficult. It is important to provide specific details about the age dispersion to maintain transparency. It's concerning that 139 cases lack mention of lesions or histology. Could you please provide information about their pregnancy and childbirth history?　 Whether or not age alone creates bias in data from a single center is not clear; exclusion criteria inconsistencies exist, such as allowing ovarian cancer but excluding cervical cancer.

Reviewer #5: The channel attention module is similar with the scale-aware distilled decoder proposed in ''COINet: Adaptive Segmentation with Co-Interactive Network for Autonomous Driving''. This paper should be cited.

Liu, Jie, et al. "COINet: Adaptive Segmentation with Co-Interactive Network for Autonomous Driving." 2021 IEEE/RSJ International Conference on Intelligent Robots and Systems (IROS). IEEE, 2021.

Reviewer #6: The paper a framework for endometrial cancer using deep learning, addressing the challenge of analyzing cytopathology images with different staining styles. The framework includes a novel CM-UNet for cell clump segmentation and an ECRNet classification algorithm based on contrastive learning.

1. The contribution of proposed work is missing. The authors need to highlight the research gaps and should explicitly mention how proposed framework filled those gaps.

2. In cytopathological images, cells can exhibit diverse shapes and sizes, making it difficult for an algorithm to generalize well across different cell types and imaging conditions. How the proposed method addresses this problem?

3. Please include one figure that illustrates the overall flow of the framework. From input image to segmentation output.

4. Please include high resolution images

5. Since the framework is solving the segmentation problem, therefore, following references need to discussed in the revised manuscript.

a. An Encoder–Decoder Deep Learning Framework for Building Footprints Extraction from Aerial Imagery. Arabian Journal for Science and Engineering, pp.1-12.

6. How many parameters are used to train the network?

7. The details of loss function are missing.

8. Experiment section is very weak. The authors need to provide more quantitative and qualitative evaluation of the framework as well as comparisons with other reference methods.

9. Please discuss failure cases and also provide their justifications

7. PLOS authors have the option to publish the peer review history of their article (what does this mean?). If published, this will include your full peer review and any attached files.

Reviewer #1: No

Reviewer #4: No

Reviewer #5: No

Reviewer #6: No

---

## [Author Response · Author response to Decision Letter 1]

19 Apr 2024

Dear Editor and Reviewers, 

Thank you very much for your careful review of our manuscript " A Deep Learning Framework for Predicting Endometrial Cancer from Cytopathologic Images with Different Staining Styles" (ID: PONE-D-23-24291). We have revised the manuscript thoroughly according to your comments and suggestions. These main changes in the revised manuscript are highlighted in red. The revisions for grammatical errors and improved sentences are marked in blue. Detailed responses to your comments are provided below.

P.S.: Citations are numbered sequentially in the order in which they appear in the text. Subject to formatting and typographical constraints, etc., a more detailed response can be found in the attached attachment.

Sincerely,

Dexing Zhong

---

## [Decision Letter · Decision Letter 2]

21 May 2024

PONE-D-23-24291R2A Deep Learning Framework for Predicting Endometrial Cancer from Cytopathologic Images with Different Staining StylesPLOS ONE

Dear Dr. Wang,

Thank you for submitting your manuscript to PLOS ONE. After careful consideration, we feel that it has merit but does not fully meet PLOS ONE’s publication criteria as it currently stands. Therefore, we invite you to submit a revised version of the manuscript that addresses the points raised during the review process.

We look forward to receiving your revised manuscript.

Kind regards,

Kazunori Nagasaka

Academic Editor

PLOS ONE

Additional Editor Comments:

Dear Authors,

Please reply to Reviwers 4 comment.

Sincerely,

Kazunori Nagasaka

Reviewers' comments:

Reviewer's Responses to Questions

**Comments to the Author**

1. If the authors have adequately addressed your comments raised in a previous round of review and you feel that this manuscript is now acceptable for publication, you may indicate that here to bypass the “Comments to the Author” section, enter your conflict of interest statement in the “Confidential to Editor” section, and submit your "Accept" recommendation.

Reviewer #4: (No Response)

Reviewer #5: All comments have been addressed

2. Is the manuscript technically sound, and do the data support the conclusions?

Reviewer #4: No

Reviewer #5: Yes

3. Has the statistical analysis been performed appropriately and rigorously? 

Reviewer #4: N/A

Reviewer #5: Yes

4. Have the authors made all data underlying the findings in their manuscript fully available?

Reviewer #4: No

Reviewer #5: Yes

5. Is the manuscript presented in an intelligible fashion and written in standard English?

Reviewer #4: (No Response)

Reviewer #5: Yes

6. Review Comments to the Author

Reviewer #4: The author was unable to respond to the comments made by reviewer #4. In 2024, medical machine learning research with a small number of data as teachers and without external validation should not be considered as a research paper, or at the very least, should not be published in PLOS One.

Reviewer #5: (No Response)

7. PLOS authors have the option to publish the peer review history of their article (what does this mean?). If published, this will include your full peer review and any attached files.

Reviewer #4: No

Reviewer #5: No

---

## [Author Response · Author response to Decision Letter 2]

4 Jun 2024

Dear Editor and Reviewers, 

Thank you very much for your careful review of our manuscript " A Deep Learning Framework for Predicting Endometrial Cancer from Cytopathologic Images with Different Staining Styles" (ID: PONE-D-23-24291). We have revised the manuscript thoroughly according to your comments and suggestions. These main changes in the revised manuscript are highlighted in red. The revisions for grammatical errors and improved sentences are marked in blue. Detailed responses to your comments are provided below.

P.S.: Citations are numbered sequentially in the order in which they appear in the text. 

Sincerely,

Dexing Zhong

 

Response to Reviewer #4:

The author was unable to respond to the comments made by reviewer #4. In 2024, medical machine learning research with a small number of data as teachers and without external validation should not be considered as a research paper, or at the very least, should not be published in PLOS One.

Responses:

Thank you for your suggestion. In response, we conducted an external validation of the paper's algorithm using the public dataset. The externally validated data were obtained from the public data platform, AIstudio, accessible via the Internet (https://aistudio.baidu.com/datasetdetail/273988). It is worth noting that the data used for external validation came from the public dataset and did not contain segmentation labels, so this external validation mainly evaluated the classification performance of ECRNet.

The dataset used for external validation consisted of 848 negative endometrial cytopathology images and 785 positive endometrial cytopathology images. The ratio of negative to positive images was 1.08:1. All images are papanicolaou stained images. During this external validation, we also chose four classification models (the ResNet-101 model, the DenseNet-121 model, the EfficientNet-B7 model, and the ResNeXt-101 model) to compare with our model. This is because these four models perform well in the classification task, second only to ECRNet. Specifically, these models above were first trained using images (1024 × 1024 pixels) from the XJTU-EC dataset instead of cell clumps (ROIs), and then tested directly on an external dataset. The specific results are shown in Table 6 as following:

TABLE 6. EXTERNAL VALIDATION COMPARISON RESULTS

Model Accuracy (%) Precision (%) Recall (%) F1-score (%)

ResNeXt-101 64.50 86.20 44.20 58.44

EfficientNet-B7 83.53 78.72 59.2 67.58

ResNet-101 73.50 100.00 53.10 69.37

DenseNet-121 80.10 77.20 67.20 71.90

Ours 95.32 94.57 96.17 95.37

In this external validation experiment, ResNet-101 has the highest precision of 100%, which means it has no false positives in the external validation. However, both ResNeXt-101, EfficientNet-B7 ResNet-101, and DenseNet-121 have low recall, which means they can easily miss screening positive patients. This is unacceptable for clinical tasks. In contrast, our model achieved the highest recall of 96.17%. In addition, ECRNet has the highest accuracy of 95.32%, followed by the DenseNet classifier with 83.53%. It is worth noting that ECRNet significantly outperforms the other four classifiers in terms of the F1 score. The F1 score is the reconciled average of precision and recall, which combines the information of precision and recall to provide a more comprehensive assessment of the performance of the classifiers, and the higher the F1 score, the better the performance of the classifiers. The higher the F1 score, the better the performance of the classifier. In summary, ECRNet has the best performance in this external validation experiment. The detailed changes (highlighted in red color) can be seen in Experiments and Results in the revised manuscript.

Finally, regarding the dataset of this manuscript, we are the first endometrial cytopathology dataset that contains segmentation and classification labels. Since making segmentation and classification labels is a laborious and time-consuming process, expanding the dataset will take more time, and we will definitely refine the information in the future. To the best of our knowledge, Koriakina et al. [1] proposed a deep multi-instance learning-based method for oral cancer detection, with data collected from a total of 24 patients; Guha et al. [2] collected 135 patients for a machine learning study of the liver; and Alhassan [3] proposed a deep learning-based method for breast cancer classification, trained and tested on a publicly available dataset of a total of 82 patients, with no external validation. 

Previously, there was a lack of deep learning-based research on endometrial cancer cytopathology images, and our work is an initial exploration in automated endometrial cytopathology screening. In addition, our model can overcome the challenge of staining bias in clinical images and has been validated in external data with some significance.

Once again, we would like to express our gratitude for your understanding and sincerely hope for your approval.

References:

[1] Koriakina N, Sladoje N, Bašić V, et al. Deep multiple instance learning versus conventional deep single instance learning for interpretable oral cancer detection[J]. Plos one, 2024, 19(4): e0302169.

[2] Guha S, Ibrahim A, Wu Q, et al. Machine learning-based identification of contrast-enhancement phase of computed tomography scans[J]. Plos one, 2024, 19(2): e0294581.

[3] Alhassan A M. An improved breast cancer classification with hybrid chaotic sand cat and Remora Optimization feature selection algorithm[J]. Plos one, 2024, 19(4): e0300622.

---

## [Decision Letter · Decision Letter 3]

11 Jun 2024

PONE-D-23-24291R3A Deep Learning Framework for Predicting Endometrial Cancer from Cytopathologic Images with Different Staining StylesPLOS ONE

Dear Dr. Wang,

Thank you for submitting your manuscript to PLOS ONE. After careful consideration, we feel that it has merit but does not fully meet PLOS ONE’s publication criteria as it currently stands. Therefore, we invite you to submit a revised version of the manuscript that addresses the points raised during the review process.

We look forward to receiving your revised manuscript.

Kind regards,

Kazunori Nagasaka

Academic Editor

PLOS ONE

Journal Requirements:

Additional Editor Comments:

Dear Authors,

Thank you for your submission.

I have received very important suggestions from one of our reviewers.

Please consider their comments and submit your revised manuscript as far as you can.

Sincerely,

Kazunori Nagasaka

Reviewers' comments:

Reviewer's Responses to Questions

**Comments to the Author**

1. If the authors have adequately addressed your comments raised in a previous round of review and you feel that this manuscript is now acceptable for publication, you may indicate that here to bypass the “Comments to the Author” section, enter your conflict of interest statement in the “Confidential to Editor” section, and submit your "Accept" recommendation.

Reviewer #4: (No Response)

2. Is the manuscript technically sound, and do the data support the conclusions?

Reviewer #4: Partly

3. Has the statistical analysis been performed appropriately and rigorously? 

Reviewer #4: N/A

4. Have the authors made all data underlying the findings in their manuscript fully available?

Reviewer #4: No

5. Is the manuscript presented in an intelligible fashion and written in standard English?

Reviewer #4: Yes

6. Review Comments to the Author

Reviewer #4: Thank you for your kind response to my repeated requests for revisions. I believe the results of the external validation are good enough to publish a paper on this AI and inquire about it with society. However, I would like you to add one more reference (reference URL) to the newly added sentence in line 438, which mentions "AIstudio, accessible via the Internet." Please include the URL along with the date of access, in the format of a literature citation. If this is completed, I recommend accepting this paper.

7. PLOS authors have the option to publish the peer review history of their article (what does this mean?). If published, this will include your full peer review and any attached files.

Reviewer #4: No

---

## [Author Response · Author response to Decision Letter 3]

11 Jun 2024

Response to Reviewer #4:

Thank you for your kind response to my repeated requests for revisions. I believe the results of the external validation are good enough to publish a paper on this AI and inquire about it with society. However, I would like you to add one more reference (reference URL) to the newly added sentence in line 438, which mentions "AIstudio, accessible via the Internet." Please include the URL along with the date of access, in the format of a literature citation. If this is completed, I recommend accepting this paper.

Responses:

Thank you for your kind suggestion. In response, we have added the reference URL [69] to the newly added sentence in line 438, referencing the example format provided by PLOS One. Specific changes (highlighted in red) can be found in the revised manuscript and in the following annexes. Finally, we would like to thank you or providing us with valuable revision opportunities to improve the quality of our paper.

Experiments and Results:

In addition, we conducted an external validation of the paper's algorithm using the public dataset. The externally validated data were obtained from the public data platform, AIstudio, accessible via the Internet [69]. It is worth noting that the data used for external validation came from the public dataset and did not contain segmentation labels, so this external validation mainly evaluated the classification performance of ECRNet.

References:

69. Endometrial Cancer; 2024 [cited 2024 Jun 4]. Database: AIstudio [Internet]. Available from: https://aistudio.baidu.com/datasetdetail/273988.

---

## [Decision Letter · Decision Letter 4]

19 Jun 2024

A Deep Learning Framework for Predicting Endometrial Cancer from Cytopathologic Images with Different Staining Styles

PONE-D-23-24291R4

Dear Dr. Wang,

We’re pleased to inform you that your manuscript has been judged scientifically suitable for publication and will be formally accepted for publication once it meets all outstanding technical requirements.

Kind regards,

Kazunori Nagasaka

Academic Editor

PLOS ONE

Additional Editor Comments (optional):

Dear Authors,

Thank you so much for your submission to Plos One.

I am pleased to tell you that your manuscirpt is acceptable for publication to our journal.

I think the manuscript is very significant and useful in the research area.

We look forward to your future manuscript.

Sincerely,

Kazunori Nagasaka

Reviewers' comments:

Reviewer's Responses to Questions

**Comments to the Author**

1. If the authors have adequately addressed your comments raised in a previous round of review and you feel that this manuscript is now acceptable for publication, you may indicate that here to bypass the “Comments to the Author” section, enter your conflict of interest statement in the “Confidential to Editor” section, and submit your "Accept" recommendation.

Reviewer #4: All comments have been addressed

2. Is the manuscript technically sound, and do the data support the conclusions?

Reviewer #4: (No Response)

3. Has the statistical analysis been performed appropriately and rigorously? 

Reviewer #4: (No Response)

4. Have the authors made all data underlying the findings in their manuscript fully available?

Reviewer #4: (No Response)

5. Is the manuscript presented in an intelligible fashion and written in standard English?

Reviewer #4: (No Response)

6. Review Comments to the Author

Reviewer #4: (No Response)

7. PLOS authors have the option to publish the peer review history of their article (what does this mean?). If published, this will include your full peer review and any attached files.

Reviewer #4: No

---

## [Editor Report · Acceptance letter]

25 Jun 2024

PONE-D-23-24291R4 

PLOS ONE

Dear Dr. Wang, 

I'm pleased to inform you that your manuscript has been deemed suitable for publication in PLOS ONE. Congratulations! Your manuscript is now being handed over to our production team.

Kind regards, 

on behalf of

Professor Kazunori Nagasaka 

Academic Editor

PLOS ONE